# Global seroprevalence of SARS-CoV-2 antibodies: A systematic review and meta-analysis

Niklas Bobrovitz[1,2‡]*, Rahul Krishan Arora[3,4‡], Christian Cao[5], Emily Boucher[5], Michael Liu[6], Claire Donnici[5], Mercedes Yanes-Lane[7], Mairead Whelan[5], Sara Perlman-Arrow[8], Judy Chen[9], Hannah Rahim[5], Natasha Ilincic[1], Mitchell Segal[1], Nathan Duarte[10], Jordan Van Wyk[10], Tingting Yan[1], Austin Atmaja[10], Simona Rocco[10], Abel Joseph[10], Lucas Penny[1], David A. Clifton[2], Tyler Williamson[4], Cedric P. Yansouni[11,12], Timothy Grant Evans[8], Jonathan Chevrier[13], Jesse Papenburg[14‡], Matthew P. Cheng[12‡]

**1** Temerty Faculty of Medicine, University of Toronto, Toronto, Ontario, Canada, **2** Department of Critical Care Medicine, University of Calgary, Calgary, Alberta, Canada, **3** Institute of Biomedical Engineering, Department of Engineering Science, University of Oxford, Oxford, United Kingdom, **4** Department of Community Health Sciences, University of Calgary, Calgary, Alberta, Canada, **5** Cumming School of Medicine, University of Calgary, Calgary, Alberta, Canada, **6** Department of Social Policy and Intervention, University of Oxford, Oxford, United Kingdom, **7** COVID-19 Immunity Task Force, McGill University, Montreal, Quebec, Canada, **8** School of Population and Global Health, McGill University, Montreal, Quebec, Canada, **9** Faculty of Medicine and Health Sciences, McGill University, Montreal, Quebec, Canada, **10** Faculty of Engineering, University of Waterloo, Waterloo, Ontario, Canada, **11** JD MacLean Centre for Tropical Diseases, McGill University, Montreal, Quebec, Canada, **12** Divisions of Infectious Diseases and Medical Microbiology, McGill University Health Centre, Montreal, Quebec, Canada, **13** Department of Epidemiology, Biostatistics and Occupational Health, Faculty of Medicine, McGill University, Montreal, Quebec, Canada, **14** Division of Pediatric Infectious Diseases, Department of Pediatrics, McGill University Health Centre, Montreal, Quebec, Canada

‡ NB and RKA contributed equally to this work and served as co-first authors. JP and MPC also contributed equally to this work and served as co-senior authors.
* niklas.bobrovitz@mail.utoronto.ca

**Data Availability Statement:** All files are available from the SeroTracker database (https://serotracker.com/). Data are from the 'Data' page of the site.

## Abstract

### Background

Many studies report the seroprevalence of severe acute respiratory syndrome coronavirus 2 (SARS-CoV-2) antibodies. We aimed to synthesize seroprevalence data to better estimate the level and distribution of SARS-CoV-2 infection, identify high-risk groups, and inform public health decision making.

### Methods

In this systematic review and meta-analysis, we searched publication databases, preprint servers, and grey literature sources for seroepidemiological study reports, from January 1, 2020 to December 31, 2020. We included studies that reported a sample size, study date, location, and seroprevalence estimate. We corrected estimates for imperfect test accuracy with Bayesian measurement error models, conducted meta-analysis to identify demographic differences in the prevalence of SARS-CoV-2 antibodies, and meta-regression to

The authors did not receive special access privileges to the data that others would not have.

**Funding:** This research was funded by the Public Health Agency of Canada through Canada's COVID-19 Immunity Task Force (https://www.covid19immunitytaskforce.ca/). DAC reports personal fees from Oxford University Innovation, Biobeats (https://www.bio-beat.com/), and Sensyne Health. MPC reports grants from McGill Interdisciplinary Initiative in Infection and Immunity and grants from Canadian Institutes of Health Research during the conduct of the study. The funders had no role in study design, data collection and analysis, decision to publish, or preparation of the manuscript.

**Competing interests:** MPC reports personal fees from Gen1E Lifesciences (as a member of the scientific advisory board) and personal fees from nplex biosciences (as a member of the scientific advisory board), both outside the submitted work. JP reports grants and personal fees from Seegene and AbbVie, grants from MedImmune and Sanofi Pasteur, outside the submitted work. RKA, NB, and TY report grants from the World Health Organization and the Canadian Medical Association for SARS-CoV-2 serosurveillance, both outside the submitted work. DAC reports personal fees from Biobeats (https://www.bio-beat.com/), and Sensyne Health during the conduct of the study. There are no patents, products in development or marketed products associated with this research to declare. This does not alter our adherence to PLOS ONE policies on sharing data and materials.

identify study-level factors associated with seroprevalence. We compared region-specific seroprevalence data to confirmed cumulative incidence. PROSPERO: CRD42020183634.

## Results

We identified 968 seroprevalence studies including 9.3 million participants in 74 countries. There were 472 studies (49%) at low or moderate risk of bias. Seroprevalence was low in the general population (median 4.5%, IQR 2.4–8.4%); however, it varied widely in specific populations from low (0.6% perinatal) to high (59% persons in assisted living and long-term care facilities). Median seroprevalence also varied by Global Burden of Disease region, from 0.6% in Southeast Asia, East Asia and Oceania to 19.5% in Sub-Saharan Africa (p<0.001). National studies had lower seroprevalence estimates than regional and local studies (p<0.001). Compared to Caucasian persons, Black persons (prevalence ratio [RR] 3.37, 95% CI 2.64–4.29), Asian persons (RR 2.47, 95% CI 1.96–3.11), Indigenous persons (RR 5.47, 95% CI 1.01–32.6), and multi-racial persons (RR 1.89, 95% CI 1.60–2.24) were more likely to be seropositive. Seroprevalence was higher among people ages 18–64 compared to 65 and over (RR 1.27, 95% CI 1.11–1.45). Health care workers in contact with infected persons had a 2.10 times (95% CI 1.28–3.44) higher risk compared to health care workers without known contact. There was no difference in seroprevalence between sex groups. Seroprevalence estimates from national studies were a median 18.1 times (IQR 5.9–38.7) higher than the corresponding SARS-CoV-2 cumulative incidence, but there was large variation between Global Burden of Disease regions from 6.7 in South Asia to 602.5 in Sub-Saharan Africa. Notable methodological limitations of serosurveys included absent reporting of test information, no statistical correction for demographics or test sensitivity and specificity, use of non-probability sampling and use of non-representative sample frames.

## Discussion

Most of the population remains susceptible to SARS-CoV-2 infection. Public health measures must be improved to protect disproportionately affected groups, including racial and ethnic minorities, until vaccine-derived herd immunity is achieved. Improvements in serosurvey design and reporting are needed for ongoing monitoring of infection prevalence and the pandemic response.

## Introduction

Over one year has passed since the World Health Organization announced on January 30, 2020 that COVID-19 was a public health emergency of international concern, yet many questions persist about the spread and impact of the virus driving this crisis [1]. As of May 15, 2021, there were over 160 million confirmed cases of SARS-CoV-2 infection and 3.3 million deaths worldwide [2]. However, these case counts inevitably underestimate the true cumulative incidence of infection [3] because of limited diagnostic test availability [4], barriers to testing accessibility [5], and asymptomatic infections [6]. As a consequence, the global prevalence of SARS-CoV-2 infection remains unknown.

Serological assays identify SARS-CoV-2 antibodies, indicating previous infection in unvaccinated persons [7]. Population-based serological testing provides better estimates of the

cumulative incidence of infection by complementing diagnostic testing of acute infection and helping to inform the public health response to COVID-19. Furthermore, as the world moves through the vaccine and variant era, synthesizing seroepidemiology findings is increasingly important to track the spread of infection, identify disproportionately affected groups, and measure progress towards herd immunity.

SARS-CoV-2 seroprevalence estimates are reported not only in published articles and pre-prints, but also in government and health institute reports, and media [8]. Consequently, few studies have comprehensively synthesized seroprevalence findings that include all of these sources [9, 10]. Describing and evaluating the characteristics of seroprevalence studies conducted over the first year of the pandemic may provide valuable guidance for serosurvey investigators moving forward.

We conducted a systematic review and meta-analysis of SARS-CoV-2 seroprevalence studies published in 2020. We aimed to: (i) describe the global prevalence of SARS-CoV-2 antibodies based on serosurveys; (ii) detect variations in seroprevalence arising from study design and geographic factors; (iii) identify populations at high risk for SARS-CoV-2 infection; and (iv) evaluate the extent to which surveillance based on detection of acute infection underestimates the spread of the pandemic.

## Methods

### Data sources and searches

This systematic review and meta-analysis was registered with PROSPERO (CRD42020183634), reported per PRISMA guidelines [11] (S1 File in S1 Materials), and will be regularly updated on an open-access platform (SeroTracker.com) [12].

We searched Medline, EMBASE, Web of Science, and Europe PMC, using a search strategy developed in consultation with a health sciences librarian (DL). The strategies for MEDLINE and EMBASE were an expanded version of the published COVID-19 search strategies created by OVID librarians for these databases [13]. Search terms related to serologic testing were identified by infectious disease specialists (MC, CY, and JP) [7] and expanded using Medical Subject Heading (MeSH) or Emtree thesauri. These searches were adapted for the other databases. The full search strategy can be found in S2 File in S1 Materials.

Given that many serosurveys are not reported in these databases [8] we used four additional search approaches to identify serosurveys reported in the grey literature. First, we searched for reports from national and international health agencies using their website search functions and examining their recurring COVID-19 reports (World Health Organization, European Centres for Disease Control, Centres for Disease Control, National Institutes of Health). Second, we searched Google News for reports of seroprevalence studies. When we encountered reports of potentially eligible government, non-governmental organizations (NGO), or academic studies, we conducted a targeted Google search to locate and include the full study. Updates of routinely reported NGO and government studies (e.g., Public Health England's weekly COVID-19 serosurveillance reports) were screened after the date they first appeared in the Google News search. Third, we consulted with international experts via e-mail to identify additional literature after all other sources had been searched. Fourth, we invited submission of seroprevalence study results on our live dashboard—SeroTracker.com.

Our search dates were from January 1, 2020 to December 31, 2020. MedRxiv pre-print articles that were updated or published as peer-review articles between January 1, 2021 and February 28, 2021, according to the MedRxiv website, were also included. No restrictions on language were applied.

## Study selection

We included SARS-CoV-2 serosurveys in humans. We defined a single serosurvey as the serological testing of a defined population over a specified time period to estimate the prevalence of SARS-CoV-2 antibodies [14, 15]. To be included, studies had to report a sample size, sampling date, geographic location of sampling, and prevalence estimate. Articles not in English or French were included if they could be fully extracted using machine translation [16]. Articles that provided information on two or more distinct cohorts (different sample frames or different samples at different time points) without a pooled estimate were considered to be multiple studies.

If multiple articles provided unique information about a study, both were included. Articles reporting identical information to previously included articles were excluded as duplicates–this rule extended to pre-print articles that were subsequently published are peer-reviewed journals. In these cases, the peer-reviewed articles were considered the definitive version.

We excluded studies conducted only in people previously diagnosed with COVID-19 using PCR, antigen testing, clinical assessment, or self-assessment; dashboards that were not associated with a defined serology study; and case reports, case-control studies, randomized controlled trials, and reviews.

## Data extraction and quality assessment

Two authors independently screened articles. Data were extracted by one reviewer and verified by a second. We extracted characteristics of the study, sample, antibody test, and seroprevalence. We extracted sub-group seroprevalence estimates when they were stratified by one variable (e.g., age) but not two variables (e.g., age and sex). Antibody isotype and time period were not considered as stratifying variables. We contacted study authors to request missing sub-group seroprevalence data.

A modified Joanna Briggs Institute (JBI) Critical Appraisal Checklist for Prevalence Studies was used to assess study risk of bias [17]. Studies were classified by overall risk of bias: low, moderate, high, or unclear (detailed criteria in S3 File in S1 Materials).

## Data synthesis and analysis

**Evaluation of seroprevalence studies and estimates.** The intended geographic scope of each estimate was classified as (A) national; (B) regional (e.g., province-level); (C) local (e.g., county-level, city-level); or (D) sublocal (e.g., one hospital department). Countries were classified according to Global Burden of Disease (GBD) region, and country income status classified by distinguishing the high-income GBD region from other regions [18, 19].

Seroprevalence studies were grouped as providing either population-wide or population-specific estimates. Population-wide studies included those using household or community sampling frames as well as convenience samples from blood donors or residual sera used for monitoring other conditions in the population. Population-specific studies were those sampling from well-defined population sub-groups, such as health care workers or long-term care residents.

We prioritized estimates based on more accurate laboratory-based assays (e.g. ELISA, CLIA), as opposed to rapid diagnostic tests. We also prioritized estimates based on IgG and anti-spike antibodies, as non-IgG and anti-nucleocapsid antibodies appear to decline more rapidly than anti-spike/RBD IgG antibodies [20–25].

Data processing and descriptive statistics were conducted in Python. *p*-values less than 0.05 were considered statistically significant.

**Correcting seroprevalence estimates.** To account for imperfect test sensitivity and specificity, seroprevalence estimates were corrected using Bayesian measurement error models, with binomial sensitivity and specificity distributions [26]. The sensitivity and specificity values for correction were derived, in order of preference, from: (i) the FindDx -McGill database of independent evaluations of serological tests [27]; (ii) independent test evaluations conducted by serosurvey investigators and reported alongside serosurvey findings; (iii) manufacturer-reported sensitivity and specificity (including author evaluated in-house assays); (iv) published pooled sensitivity and specificity by immunoassay type [25]. If uncorrected estimates were not available, we used author-reported corrected seroprevalence estimates. Details of these evaluations are located in S4 File in S1 Materials.

We presented corrected and uncorrected estimates for all studies. Subsequent analyses were done using corrected seroprevalence estimates. To assess the impact of correction, we calculated the absolute difference between seroprevalence estimates before and after correction. We also conducted each analysis with uncorrected data.

**Global seroprevalence and associated factors.** To examine study-level factors affecting population-wide seroprevalence estimates, we constructed a multivariable linear meta-regression model. The outcome variable was the natural logarithm of corrected seroprevalence. Independent predictors were defined *a priori*. Categorical covariates were encoded as indicator variables, and included: study risk of bias (reference: low risk of bias), GBD region (reference: high-income); geographic scope (reference: national); and population sampled (reference: household and community samples). The sole continuous covariate was the cumulative number of confirmed cases in the country of the study. We obtained data on total confirmed SARS-CoV-2 infections [28, 29] and population size [30] that geographically matched the study populations nine days before the study end date, to reflect the time period between COVID-19 diagnosis and seroconversion (S5 File in S1 Materials) [31–33]. A quantile-quantile plot and a funnel plot were generated to visually check normality and homoscedasticity. All meta-analysis and meta-regression were done using the meta package in R [34].

**Population differences in seroprevalence.** To quantify population differences in SARS-CoV-2 seroprevalence, we identified subgroup estimates within population-wide studies that stratified by sex/gender, race/ethnicity, contact with individuals with COVID-19, occupation, and age groups. We calculated the ratio in prevalence between groups within each study (e.g., prevalence in males vs. females) then aggregated the ratios across studies using inverse variance-weighted random-effects meta-analysis (S4 File in S1 Materials). Heterogeneity was quantified using the $I^2$ statistic [35].

**Comparisons of seroprevalence and confirmed SARS-CoV-2 infections.** To measure how much confirmed SARS-CoV-2 infections detected using RT-PCR underestimate seroprevalence, we calculated the ratio between population-wide seroprevalence estimates and the cumulative incidence of confirmed SARS-CoV-2 infections.

## Results

### Characteristics of included studies

We screened 24,999 titles and abstracts and 1,830 full text articles (Fig 1). We identified 968 unique seroprevalence studies in 605 articles. These studies included 9,329,185 participants.

There were 590 (61%) population-wide studies and 378 (39%) population-specific studies (Table 1). Characteristics of individual studies are reported in S1 and S2 Tables in S1 Materials. Study sampling dates ranged from September 1, 2019 to December 31, 2020.

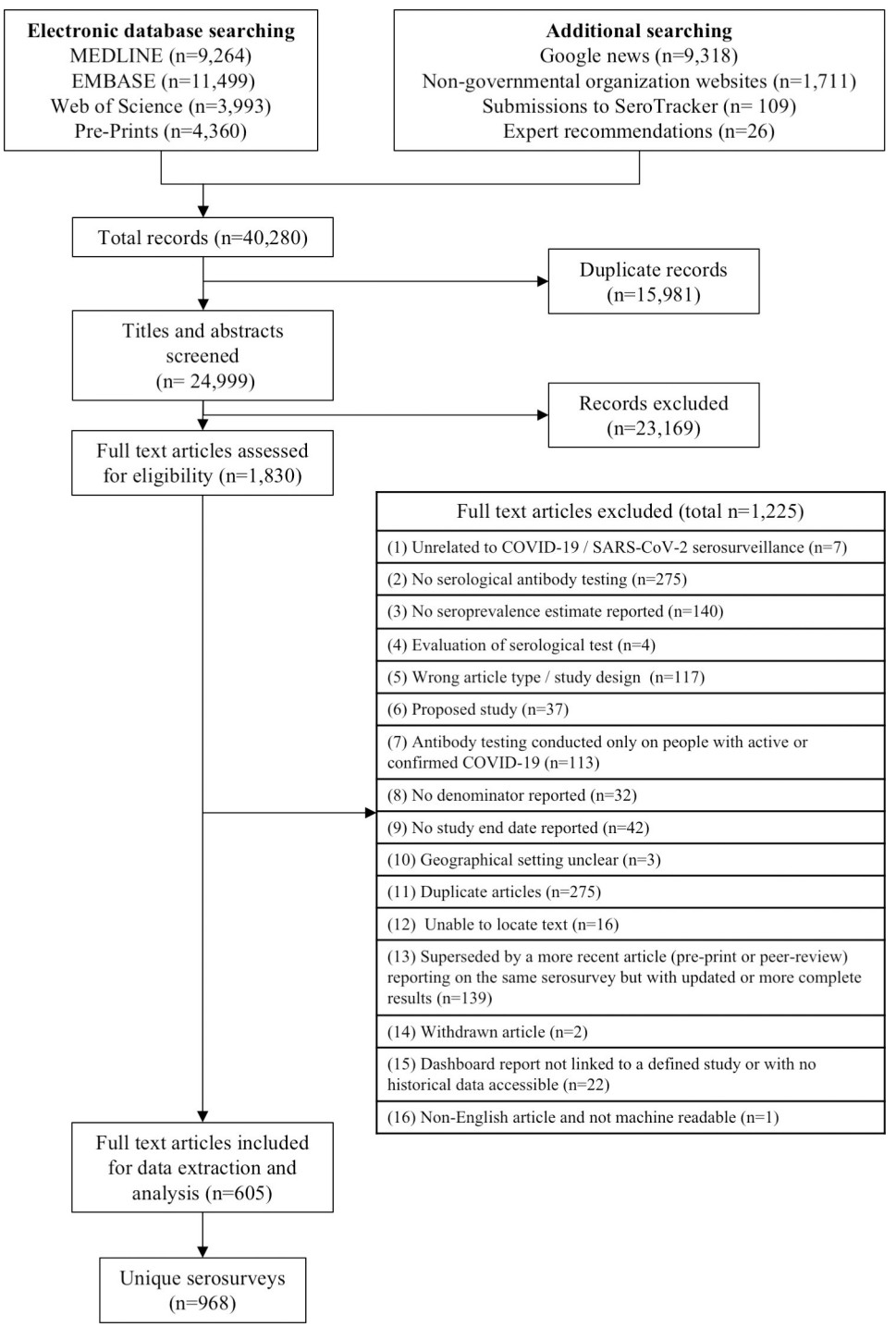

**Fig 1. PRISMA flow diagram of study inclusion.**

Seventy-four countries across all GBD regions were represented among identified serosurveys (Fig 2; S1 Fig in S1 Materials). A minority of studies were conducted in low- and middle-income countries (n = 221, 23%).

**Table 1. Summary characteristics of included articles.**

| Characteristic | Studies n (%) |
|---|---|
| **Geographic scope** | |
| National | 116 (12%) |
| Regional | 347 (36%) |
| Local | 277 (29%) |
| Sublocal | 228 (24%) |
| **Age groups[a]** | |
| Children and Youth (0–17 years) | 28 (3%) |
| Adults (18–64 years) | 268 (28%) |
| Seniors (65+ years) | 7 (0.7%) |
| Multiple age groups | 609 (63%) |
| **Population** | |
| Studies reporting population-wide estimates | 590 (61%) |
| Studiesreporting population-specific estimates[b] | 378 (39%) |
| **County income level[c]** | |
| High income | 747 (77%) |
| Low/middle income | 221 (23%) |
| **Sampling method** | |
| Probability sampling | 209 (22%) |
| Non-probability sampling | 759 (78%) |
| **Antibody tests[d]** | |
| ELISA | 242 (25%) |
| CLIA | 409 (42%) |
| LFIA | 137 (14%) |
| Other | 10 (1%) |
| Neutralization | 4 (0.4%) |
| Multiple types | 37 (4%) |
| **Antibody isotypes reported[d]** | |
| IgG | 845 (87%) |
| IgM | 227 (24%) |
| IgA | 47 (5%) |
| **Risk of bias** | |
| Low | 28 (3%) |
| Moderate | 443 (46%) |
| High | 424 (44%) |
| Unclear | 73 (8%) |

[a]When the age range for participants in a study overlapped multiple age categories by $> = 30\%$ then the study was counted as examining multiple age groups.

[b]Studies sampling from well-defined population sub-groups.

[c]Classified according to the WHO global burden of disease region groupings (high vs other—low/middle).

[d]Studies could have met multiple criteria so the sum of percentages may exceed 100%. Abbreviations: ELISA = enzyme-linked immunosorbent assay; CLIA = chemiluminescence immunoassay; LFIA = lateral flow immunoassay.

Many studies were at moderate (n = 443, 46%) or high risk of bias (n = 424, 44%), owing primarily to the absence of statistical correction either for population demographics or test sensitivity and specificity, using non-probability sampling methods, and using non-representative sample frames (Fig 3, S3 Table in S1 Materials).

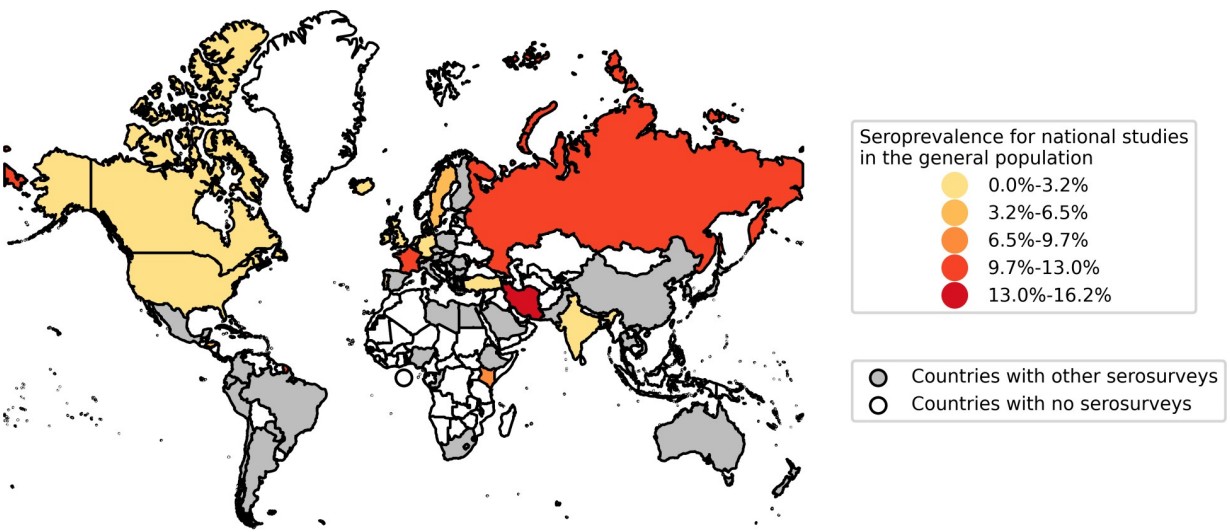

**Fig 2. Map of national seroprevalence studies reporting population-wide estimates.** Countries with national-level seroprevalence studies reporting population-wide estimates are coloured on the map, based on the seroprevalence reported in the most recent such study in each country. Countries with no such national serosurveys but with "other serosurveys" are coloured in grey; this includes local and regional studies, as well as studies in specific populations. Map data reprinted from Natural Earth under a CC BY license, with permission from Natural Earth, original copyright 2009.

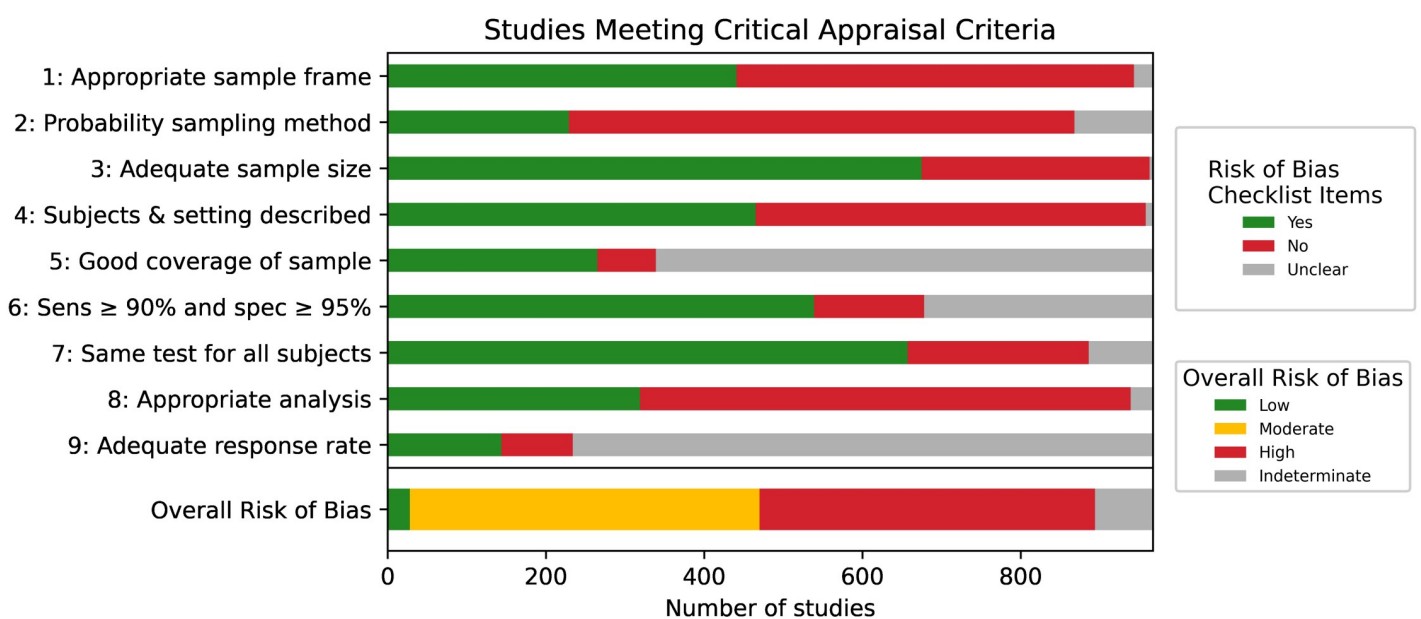

**Fig 3. Study risk of bias summary.** Item 1: Was the sample frame appropriate to address the target population? Item 2: Were study participants recruited in an appropriate way? Item 3: Was the sample size adequate? Item 4: Were the study subjects and setting described in detail? Item 5: Was data analysis conducted with sufficient coverage of the identified sample? Item 6: Were valid methods used for the identification of the condition? Item 7: Was the condition measured in a standard, reliable way for all participants? Item 8: Was there appropriate statistical analysis? Item 9: Was the response rate adequate, and if not, was the low response rate managed appropriately? Item 10: Overall risk of bias.

## Correction of estimates for test sensitivity and specificity

In order to improve comparability between data and correct for misclassification error, we corrected seroprevalence values for imperfect sensitivity and specificity. To do so, we sourced additional evaluation data as described in the methods. Overall, there were 795 studies (82%) for which test sensitivity and specificity values were reported or located (S5 Table in S1 Materials). Authors reported sensitivity and specificity data in 229 studies, with reported sensitivity values ranging from 35–100% and specificity between 87–100%.

Independent evaluation data from the FindDx initiative were available for 359 studies (37%), manufacturer evaluations were available for 182 studies (19%), and published pooled sensitivity and specificity results for ELISAs, LFIAs, and CLIAs, based on the test type known to have been used, and using the definitions for these test types provided by Bastos et al. [25], were available for 101 studies (10%). Between FindDx, manufacturer evaluations, and published pooled results, test sensitivity ranged from 9–100% and specificity from 0–100%.

Estimates from 587 studies (61%) were corrected for imperfect sensitivity and specificity. We corrected seroprevalence estimates from 290 studies (30%), while author-corrected estimates were used in 297 (31%) studies as uncorrected estimates were not available for our analysis. The median absolute difference between corrected and uncorrected seroprevalence estimates was 1.1% (IQR 0.6–2.3%).

Of the 381 studies for which estimates were not corrected, data were insufficient to inform the correction analysis in 118 studies (12%). Corrected seroprevalence estimates could not be determined for 261 studies (27%), most of which were population-specific studies using small sample sizes and low test sensitivity and specificity. In these studies, the model used to correct for test sensitivity and specificity often failed to converge to a reasonable adjusted prevalence value.

## Population-wide seroprevalence estimates

In studies reporting population-wide seroprevalence estimates, median corrected seroprevalence was 4.5% (IQR 2.4–8.4%, Table 2). These studies included household and community samples (n = 125), residual sera (n = 248), and blood donors (n = 54), with median corrected seroprevalence of 6.0% (IQR 2.8–15.1%), 4.0% (IQR 2.4–6.8%), and 4.7% (IQR 1.4–6.8%), respectively (Table 3).

Among high-income countries, the median corrected seroprevalence in studies reporting population-wide estimates 4.1% (IQR 2.4–6.9%). In the low- and middle-income GBD regions, median corrected seroprevalence ranged from 0.6% (IQR 0.3–1.4%) in Southeast Asia, East Asia, and Oceania to 19.5% (IQR 9.0–26.0%) in South Asia (Table 2).

## Population-specific seroprevalence estimates

The median corrected seroprevalence in studies reporting population-specific seroprevalence estimates was 3.6%, (IQR 0.9–12.3%, Table 4) however, there was wide variation (0.6–59%) between different populations (Table 3). Notably, the median corrected seroprevalence was 3.6% (IQR 0.8–11.0%, n = 66 studies) in healthcare workers and caregivers and 2.7% (IQR 1.1–7.4%, n = 24 studies) in specific patient groups (e.g., cancer patients). Essential non-healthcare workers (e.g., first responders) had a median seroprevalence of 7.5% (IQR 2.4–29.9%, n = 11 studies, Table 3). Higher seroprevalence estimates were reported in studies of contacts of COVID-19 patients (median 31.5%, IQR 2.7–39.9%, n = 11 studies), persons living in slums (median 41.7%, IQR 40.0–43.4%, n = 2 studies), and persons in assisted living and long-term care facilities (median 59.2%, IQR 39.7–78.8%, n = 2 studies).

**Table 2. Summary of seroprevalence data from studies reporting population-wide estimates by global burden of disease region, geographic scope, and risk of bias.**

| Characteristic | No. studies | No. countries | Median sample size (IQR) | Median uncorrected seroprevalence (IQR) | No. studies with correctable data | Median corrected seroprevalence (IQR) | Risk of bias |
|---|---|---|---|---|---|---|---|
| **Population-wide studies** | 590 | 57 | 987 (786–2639) | 4.6% (2.2–8.5%) | 427 | 4.5% (2.4–8.4%) | L: 4%, M: 62%, H: 27%, U: 6% |
| **GBD region** | | | | | | | |
| Central Europe, Eastern Europe, and Central Asia | 14 | 6 | 2681 (992–3037) | 7.8% (2.3–20.5%) | 9 | 12.2% (4.5–25.4%) | L: 7%, M: 43%, H: 43%, U: 7% |
| High-income | 453 | 28 | 985 (786–1709) | 4.4% (2.2–7.2%) | 339 | 4.1% (2.4–6.9%) | L: 3%, M: 65%, H: 27%, U: 5% |
| Latin America and Caribbean | 57 | 10 | 900 (832–1968) | 6.8% (2.6–19.5%) | 37 | 10.6% (3.0–46.5%) | L: 5%, M: 70%, H: 18%, U: 7% |
| North Africa and Middle East | 5 | 4 | 1212 (600–3530) | 12.9% (0.8–19.3%) | 4 | 8.2% (0.1–17.7%) | L: 20%, M: 40%, H: 20%, U: 20% |
| South Asia | 35 | 2 | 3000 (502–15625) | 17.6% (8.8–26.8%) | 25 | 17.1% (8.7–25.0%) | L: 17%, M: 43%, H: 20%, U: 20% |
| Southeast Asia, East Asia, and Oceania | 20 | 2 | 2192 (434–18024) | 1.0% (0.4–2.9%) | 8 | 0.6% (0.3–1.4%) | L: 0%, M: 35%, H: 60%, U: 5% |
| Sub-Saharan Africa | 6 | 5 | 528 (214–2282) | 14.6% (8.0–24.0%) | 5 | 19.5% (9.0–26.0%) | L: 17%, M: 33%, H: 50%, U: 0% |
| **Scope** | | | | | | | |
| National | 83 | 32 | 4297 (1200–24926) | 4.5% (1.9–6.1%) | 51 | 3.5% (1.2–6.0%) | L: 10%, M: 64%, H: 19%, U: 7% |
| Regional | 312 | 19 | 980 (802–1106) | 4.3% (2.3–7.6%) | 276 | 4.5% (2.5–7.6%) | L: 4%, M: 73%, H: 21%, U: 3% |
| Local | 167 | 31 | 1000 (752–2547) | 5.5% (2.1–14.8%) | 87 | 6.7% (2.6–21.9%) | L: 4%, M: 46%, H: 38%, U: 13% |
| Sub-local | 28 | 14 | 500 (357–928) | 7.2% (1.4–15.1%) | 13 | 8.7% (0.6–15.1%) | L: 0%, M: 36%, H: 61%, U: 4% |
| **Risk of bias** | | | | | | | |
| Low | 25 | 13 | 4151 (2203–9922) | 8.2% (2.9–13.6%) | 20 | 10.3% (3.3–18.9%) | .. |
| Moderate | 367 | 42 | 985 (900–1545) | 4.7% (2.6–7.9%) | 307 | 4.5% (2.5–7.9%) | .. |
| High | 161 | 30 | 731 (313–2415) | 3.9% (1.2–9.4%) | 93 | 3.9% (0.9–8.2%) | .. |
| Unclear | 37 | 15 | 1709 (774–8006) | 3.3% (1.5–11.0%) | 7 | 11.7% (4.8–24.6%) | .. |

Abbreviations: No. = number; IQR = interquartile range; L = low; M = moderate; H = high; U = unclear; GBD = global burden of disease region.

### Seroprevalence by population sub-groups (meta-analysis)

Within studies, seroprevalence was significantly lower for seniors 65+ compared to adults 18–64 (prevalence ratio [PR]: 0.79 [95% CI: 0.69–0.90]). Seroprevalence was significantly higher for Black persons, Asian persons, Indigenous persons, and other groups compared to Caucasian persons (PRs from 1.89–5.74), and in health care workers with close contact with COVID-19 patients compared to those with no close contact (PR 2.10 [1.28–3.44]). Seroprevalence differences approached significance for individuals in the community with close contact with COVID-19 patients (PR 1.85 [0.99–3.44]) and for health care workers compared to members of the community (PR 1.45 [0.99–2.14]). There were no differences in infection risk based on sex and gender. Full results are reported in Table 5, and results for uncorrected prevalence estimates are reported in S4 Table in S1 Materials.

**Table 3. Summary of seroprevalence data by study sampling frame.**

| Population | No. of studies | Median sample size (IQR) | Median uncorrected seroprevalence (IQR) | No. of studies with correctable data | Median corrected seroprevalence (IQR) | Risk of Bias |
|---|---|---|---|---|---|---|
| **Population-wide studies** | 590 | 987 (786–2639) | 4.6% (2.2–8.5%) | 427 | 4.5% (2.4–8.4%) | L: 4%, M: 62%, H: 27%, U: 6% |
| Residual sera | 289 | 980 (804–1043) | 4.1% (2.2–7.1%) | 248 | 4.0 (2.4–6.8) | L: 0%, M: 72%, H: 28%, U: 0% |
| Household and community samples | 228 | 1530 (615–4889) | 5.7% (2.4–12.0%) | 125 | 6.0 (2.8–15.1) | L: 10%, M: 49%, H: 26%, U: 14% |
| Blood donors | 73 | 1110 (881–7389) | 4.0% (1.8–10.3%) | 54 | 4.7 (1.4–11.1) | L: 1%, M: 66%, H: 29%, U: 4% |
| **Population-specific studies** | **378** | **634 (200–1694)** | **5.3% (1.7–14.0%)** | **160** | **3.6% (0.9–12.3%)** | **L: 1%, M: 20%, H: 70%, U: 10%** |
| Health care workers and caregivers | 191 | 801 (242–2420) | 5.0% (1.7–12.0%) | 66 | 3.6 (0.8–11.0) | L: 1%, M: 23%, H: 68%, U: 9% |
| Patients seeking care for non-COVID-19 reasons | 46 | 229 (94–560) | 3.6% (1.5–9.2%) | 24 | 2.7 (1.1–7.4) | L: 0%, M: 7%, H: 83%, U: 11% |
| Multiple populations | 41 | 1159 (276–4656) | 5.5% (1.5–14.8%) | 23 | 3.2 (0.3–11.3) | L: 2%, M: 17%, H: 71%, U: 10% |
| Essential non-healthcare workers | 27 | 405 (239–992) | 4.3% (2.2–14.8%) | 11 | 7.5 (2.4–29.9) | L: 0%, M: 15%, H: 78%, U: 7% |
| Contacts of COVID patients | 18 | 178 (71–302) | 17.7% (1.3–35.2%) | 11 | 31.5 (2.7–49.5) | L: 0%, M: 33%, H: 61%, U: 6% |
| Pregnant or parturient women | 17 | 433 (169–1000) | 5.8% (2.1–8.3%) | 8 | 3.7 (1.7–5.8) | L: 0%, M: 24%, H: 76%, U: 0% |
| Non-essential workers and unemployed persons | 13 | 2500 (1007–2715) | 2.6% (1.0–20.0%) | 8 | 1.5 (0.8–7.7) | L: 0%, M: 38%, H: 54%, U: 8% |
| Assisted living and long-term care facilities | 9 | 291 (150–371) | 23.6% (17.3–39.0%) | 2 | 59.2 (39.7–78.8) | L: 0%, M: 0%, H: 78%, U: 22% |
| Persons who are incarcerated | 4 | 1034 (664–1213) | 50.3% (29.3–72.2%) | 0 | - | L: 0%, M: 0%, H: 0%, U: 100% |
| Family of essential workers | 3 | 849 (484–920) | 7.7% (5.4–15.6%) | 0 | - | L: 0%, M: 33%, H: 67%, U: 0% |
| Students and day-cares | 2 | 900 (845–954) | 7.0% (5.5–8.4%) | 2 | 4.6 (4.3–4.9) | L: 0%, M: 50%, H: 50%, U: 0% |
| Persons experiencing homelessness | 2 | 474 (301–646) | 28.4% (16.5–40.2%) | 1 | 2.8 (2.8–2.8) | L: 0%, M: 0%, H: 100%, U: 0% |
| Persons living in slums | 2 | 2131 (1096–3166) | 45.0% (40.5–49.6%) | 2 | 41.7 (40.0–43.4) | L: 50%, M: 0%, H: 50%, U: 0% |
| Tissue donor | 1 | 235 (235–235) | 0.9% (0.9–0.9%) | 0 | - | L: 0%, M: 0%, H: 100%, U: 0% |
| Perinatal | 1 | 1206 (1206–1206) | 1.4% (1.4–1.4%) | 1 | 0.6 (0.6–0.6) | L: 0%, M: 0%, H: 100%, U: 0% |
| Hospital visitors | 1 | 1188 (1188–1188) | 2.7% (2.7–2.7%) | 1 | 1.5 (1.5–1.5) | L: 0%, M: 100%, H: 0%, U: 0% |

Abbreviations: No. = number; IQR = interquartile range; L = low; M = moderate; H = high; U = unclear; GBD = global burden of disease region.

## Seroprevalence by study and geographic factors (meta-regression)

On multivariable meta-regression, studies at low risk of bias reported higher corrected seroprevalence estimates relative to studies with moderate risk of bias (prevalence ratio 1.67, 95% CI 1.22–2.27, p = 0.001), high risk of bias (1.54, 95% CI 1.11–2.13, p = 0.01), and unclear risk of bias (2.63, 95% CI 1.54–4.55, p<0.001)(S6 Table in S1 Materials). Blood donors and residual sera groups, both used as proxies for the general population, reported similar corrected

**Table 4. Summary of seroprevalence data from studies reporting population-specific estimates by global burden of disease region, geographic scope, and risk of bias.**

| Characteristic | No. studies | No. countries | Median sample size (IQR) | Median uncorrected seroprevalence (IQR) | No. studies with correctable data | Median corrected seroprevalence (IQR) | Risk of bias |
|---|---|---|---|---|---|---|---|
| **Population-specific studies** | 378 | 53 | 634 (200–1694) | 5.3% (1.7–14.0%) | 160 | 3.6% (0.9–12.3%) | L: 1%, M: 20%, H: 70%, U: 10% |
| **GBD region** | | | | | | | |
| Central Europe, Eastern Europe, and Central Asia | 12 | 7 | 512 (354–1611) | 2.8% (1.2–10.7%) | 5 | 10.6% (8.8–14.4%) | L: 0%, M: 33%, H: 42%, U: 25% |
| High-income | 294 | 24 | 611 (188–1662) | 5.1% (1.8–12.1%) | 125 | 3.2% (0.9–10.0%) | L: 0%, M: 19%, H: 71%, U: 9% |
| Latin America and Caribbean | 12 | 6 | 378 (275–1820) | 9.8% (5.7–13.7%) | 7 | 10.7% (4.6–16.5%) | L: 8%, M: 25%, H: 58%, U: 8% |
| North Africa and Middle East | 16 | 7 | 434 (223–2991) | 16.8% (3.8–38.7%) | 7 | 29.4% (20.0–45.8%) | L: 0%, M: 25%, H: 75%, U: 0% |
| South Asia | 14 | 2 | 1006 (671–1537) | 16.6% (11.3–30.7%) | 2 | 28.1% (19.6–36.6%) | L: 7%, M: 29%, H: 50%, U: 14% |
| Southeast Asia, East Asia, and Oceania | 26 | 3 | 1024 (346–4418) | 1.9% (0.3–5.3%) | 13 | 0.3% (0.2–3.5%) | L: 0%, M: 19%, H: 69%, U: 12% |
| Sub-Saharan Africa | 4 | 4 | 452 (320–614) | 20.2% (12.8–24.3%) | 1 | 11.3% (11.3–11.3%) | L: 0%, M: 0%, H: 100%, U: 0% |
| **Scope** | | | | | | | |
| National | 33 | 24 | 1150 (525–4234) | 3.8% (1.7–11.6%) | 19 | 4.5% (0.5–12.1%) | L: 0%, M: 24%, H: 55%, U: 21% |
| Regional | 35 | 14 | 1671 (320–4814) | 3.1% (1.5–13.5%) | 15 | 3.7% (1.9–19.4%) | L: 3%, M: 37%, H: 57%, U: 3% |
| Local | 110 | 28 | 681 (206–1654) | 5.1% (1.9–14.4%) | 49 | 3.0% (0.8–11.5%) | L: 2%, M: 20%, H: 71%, U: 7% |
| Sub-local | 200 | 33 | 376 (174–1156) | 6.0% (1.9–14.0%) | 77 | 4.0% (0.9–12.0%) | L: 0%, M: 16%, H: 74%, U: 10% |
| **Risk of bias** | | | | | | | |
| Low | 3 | 3 | 4202 (2770–16497) | 29.1% (16.6–41.6%) | 3 | 45.1% (24.7–56.6%) | .. |
| Moderate | 76 | 27 | 1808 (922–4127) | 5.1% (2.3–11.3%) | 34 | 3.4% (1.4–8.6%) | .. |
| High | 263 | 42 | 320 (152–1002) | 5.4% (1.7–15.1%) | 113 | 3.4% (0.8–13.4%) | .. |
| Unclear | 36 | 16 | 1098 (354–2880) | 3.8% (0.9–10.0%) | 10 | 4.6% (2.7–7.4%) | .. |

Abbreviations: No. = number; IQR = interquartile range; L = low; M = moderate; H = high; U = unclear; GBD = global burden of disease region.

seroprevalence estimates compared to household and community samples (blood donors: 0.96, 95% CI 0.76–1.22, p = 0.77; residual sera: 1.12, 95% CI 0.94–1.35).

National studies reported lower seroprevalence estimates compared to regional studies (0.61, 95% CI 0.48–0.77, p<0.001), local studies (0.47, 95% CI 0.37–0.60, p<0.001) and sublocal studies (0.52, 95% CI 0.33–0.81, p = 0.004). Finally, compared to high-income countries, higher seroprevalence estimates were reported by countries in Sub-Saharan Africa (5.01, 95% CI 2.89–8.69, p<0.001), South Asia (2.84, 95% CI 2.09–3.85, p<0.001), Central Europe, Eastern Europe, and Central Asia (2.83, 95% CI 1.75–4.55, p<0.001), and Latin America and Caribbean (2.71, 95% CI 2.07–3.54, p<0.001), while countries in Southeast Asia, East Asia, and Oceania (0.18, 95% CI 0.09–0.34) reported lower seroprevalence estimates. Visual checks confirmed that model assumptions of normality and homoscedasticity were met.

**Table 5. Differences in seroprevalence estimates by demographic characteristics within studies.**

| Factor | Reference Group | Comparison Group | Number of Studies | Risk Ratio (95% CI)[a] | Heterogeneity ($I^2$) |
|---|---|---|---|---|---|
| Age | Adults (18–64) | Youth (0–17) | 82 | 0.92 (0.81–1.04) | 90.7% |
| | Adults (18–64) | Seniors (65+) | 127 | 0.79 (0.69–0.90) | 93.9% |
| Sex/Gender | Female | Male | 129 | 1.03 (0.98–1.08) | 79.1% |
| Race | Caucasian | Black | 19 | 3.37 (2.64–4.29) | 85.7% |
| | Caucasian | Asian | 17 | 2.47 (1.96–3.11) | 88.9% |
| | Caucasian | Indigenous | 8 | 5.74 (1.01–32.6) | 75.3% |
| | Caucasian | Multiple/other | 18 | 1.89 (1.60–2.24) | 64.0% |
| Close contact with COVID-19 patients | Individuals with no close contact | Individuals with close contact | 35 | 1.85 (0.99–3.44) | 97.4% |
| | Health care workers with no close contact | Health care workers with close contact | 44 | 2.10 (1.28–3.44) | 89.4% |
| Health care worker status | Non-health care workers and caregivers | Health care workers and caregivers | 19 | 1.45 (0.99–2.14) | 98.3% |

[a]Using corrected seroprevalence estimates. Abbreviations: CI = confidence interval.

### Ratio of seroprevalence to cumulative case incidence

The median ratio between corrected seroprevalence estimates from national studies and the corresponding cumulative incidence of SARS-CoV-2 infection nine days prior was 18.1 (IQR 5.9–38.7, n = 49 studies; Table 6, S2 Fig in S1 Materials), indicating a median of 18.1 serologically identified infections per 1 confirmed case globally. Stratifying by risk of bias and GBD showed variation in median ratios between seroprevalence and cumulative incidence (Table 6).

## Discussion

This systematic review and meta-analysis provides an overview of global SARS-CoV-2 seroprevalence based on data from 9,329,185 participants in 968 serosurveys from 605 reports.

**Table 6. The median ratio between corrected seroprevalence estimates from national studies and the corresponding cumulative incidence of SARS-CoV-2 infection from nine days prior.**

| Characteristics | Number of studies | Ratio of seroprevalence to cumulative incidence |
|---|---|---|
| **National studies with correctable estimates and matching case data available** | 49 | 18.1 (5.9–38.7) |
| **Risk of bias** | | |
| Low | 6 | 19.9 (11.2–111.7) |
| Moderate | 31 | 12.1 (5.3–32.9) |
| High | 10 | 19.4 (18.8–39.3) |
| Unclear | 2 | 0.4 (0.3–0.5) |
| **Global burden of disease regions** | | |
| Central Europe, Eastern Europe, and Central Asia[a] | - | - |
| High-income | 41 | 15.2 (5.9–24.2) |
| Latin America and Caribbean | 3 | 49.5 (46.7–75.7) |
| North Africa and Middle East | 2 | 71.2 (35.7–106.7) |
| South Asia | 2 | 6.7 (6.1–7.4) |
| Southeast Asia, East Asia, and Oceania[a] | - | - |
| Sub-Saharan Africa | 1 | 602.5 (602.5–602.5) |

[a]Matching cumulative incidence data not available for the seroprevalence study periods.

Overall, in the first year of the COVID-19 pandemic, estimates of population-wide seroprevalence were low (median 4.5%, IQR 2.4–8.4%), however, population-specific estimates of seroprevalence varied widely from a low of 0.6% (perinatal) to a high of 59% (persons in assisted living and long-term care facilities).

Seroprevalence varied considerably between GBD regions after correcting for study characteristics and test sensitivity and specificity. Given the limited evidence for altitude or climate effects on SARS-CoV-2 transmission [36, 37] variations in seroprevalence likely reflect differences in community transmission based on behaviour, public health responses, local resources, and the built environment. Stakeholders should carefully review the infection control measures implemented in Southeast Asia, East Asia, and Oceania as they appear to have been effective at limiting SARS-CoV-2 transmission [38, 39].

Our results suggest clear population differences in SARS-CoV-2 infection, with marginalized and high-risk groups disproportionately affected. Differences in infection risk based on race might be attributed to crowding, higher-risk occupation roles (e.g., front-line service jobs) and other systemic inequities [40–43]. Some of these groups (Black, Asian, and other minority racial and ethnic groups) are also known to have higher infection fatality rates [44]. Such differences may inform policy on vaccine distribution, workforce protections, and other public health measures designed to protect marginalized persons.

Our review found that health care workers who had close contact with confirmed COVID-19 cases had a higher risk of seropositivity, consistent with previous reports [45]. Results in this study regarding contact with a COVID-19 case among non-health care workers warrant further investigation. Our meta-analysis of seroprevalence in persons with and without contact in studies reporting both subgroups found no significant difference, despite the fact that studies of persons with exposure to COVID-19 reported much higher seroprevalence estimates compared to population-wide studies (31.5% vs. 4.5%). These results align with other evidence synthesis examining persons with and without COVID-19 exposure however, they conflict with studies of high-risk exposure, including health care workers [9, 46]. It is possible that contact exposure in a clinical setting may be more narrowly defined and carefully measured, whereas definitions of exposure in non-clinical studies may be more heterogenous or prone to potential misclassification due to asymptomatic infection. Future analysis should explore the association of different definitions and measurement of contact status with seroprevalence estimates.

Few studies (23%) have been conducted in low- and middle-income countries. Results from the ongoing WHO Unity studies will help to bridge this knowledge gap and contribute to a more comprehensive understanding of the spread and impact of COVID-19 globally [15]. Use of the standardized Unity protocols will also help to increase the pool of robust, comparable seroprevalence data.

Approximately half of studies reporting population-wide SARS-CoV-2 seroprevalence estimates used blood from donors and residual sera as a proxy for the community. Our results showed that these studies report seroprevalence estimates that are similar to studies of household and community-based samples. It has previously been shown that these groups contain disproportionate numbers of people that are young, White, college graduates, employed, physically active, and never-smokers [47, 48]. However, the results of our study suggest that investigators may use these proxy sampling frames to obtain fairly representative estimates of seroprevalence if studies use large sample sizes with adequate coverage of important subgroups (e.g., age, sex, race/ethnicity) to permit standardization to population characteristics, tests with high sensitivity and specificity, and statistical corrections for imperfect sensitivity and specificity.

Our results suggest that studies at moderate, high, or unclear risk of bias may generate lower seroprevalence estimates relative to studies at low risk of bias. There are many possible explanations for this somewhat counterintuitive finding. Common reasons for unclear or elevated risk of bias were absent reporting of test information, use of tests with low sensitivity and specificity, no statistical correction for demographics or test sensitivity and specificity, use of non-probability sampling, and use of non-representative sample frames. Therefore, selection bias that favoured healthier, affluent, non-racialised groups at lower risk of infection paired with no adjustment for sample characteristics may have contributed to lower estimates of seroprevalence. It is also possible that the false negative rate was higher for studies in which authors used low sensitivity tests, particularly when authors did not statistically correct estimates for imperfect test performance or used inflated estimates of test sensitivity, as are often reported by manufactures, to conduct such corrections.

Systematic reviews of SARS-CoV-2 serological test accuracy have found that many tests have poor sensitivity and specificity [24, 25]. Of the studies included in this review, only 298 (31%) corrected for test sensitivity and specificity, and 118 (12%) failed to report identifying information on the test used altogether. Our study corrected seroprevalence estimates for test sensitivity and specificity in an additional 290 (30%) studies. The median absolute difference between corrected and uncorrected estimates was 1.1%—a substantial change, given that the median corrected seroprevalence in studies reporting population-wide estimates was 4.5%. This difference emphasizes the importance of conducting such corrections to minimize bias in serosurvey data. Furthermore, improved reporting of serological testing information in serosurveys is needed to maximize the amount of robust and comparable data for evidence synthesis.

Seroprevalence estimates were 18.1 times higher than the corresponding cumulative incidence of COVID-19 infections, with large variations between the Global Burden of Disease Regions (seroprevalence estimates ranging from 6 to 602 times higher than cumulative incidence). This level of under-ascertainment suggests that confirmed SARS-CoV-2 infections are a poor indicator of the extent of infection spread, even in high-income countries where testing has been more widely available. The broad range of ratios mirrors estimates from other published evidence on case under-ascertainment, which suggests a range of 0.56 to 717 [49, 50].

Seroprevalence to cumulative case ratios can provide a rough roadmap for public health authorities by identifying areas that may be receiving potentially insufficient levels of testing and by providing an indication of the number of undetected asymptomatic infections.

While there is interest in using these seroprevalence to cumulative case ratios in identifying inadequate testing and estimating case ascertainment, caution is required in the quantitative interpretation of these ratios. Our study found a median ratio of 18.1, which aligns with other published analysis [50]. This would imply that 2.9 billion people globally have been infected with SARS-CoV-2 rather than the 160 million reported as of May 15, 2021 [2]. This is not likely, and this estimate conflicts with the evidence that seroprevalence remains low in the general population. If applying this global ratio to countries with high cumulative incidence, such as the United States (32 million by May 15, 2021), then the total number of infections would exceed the population.

There are several possible reasons for these discrepancies. Firstly, these ratios clearly vary by geographic region and regional health policy, with higher diagnostic testing rates likely to correspond to lower seroprevalence to case ratios. Country-specific ratios, or region-specific ratios if available, should be used to inform planning wherever possible. Second, diagnostic testing-based estimates of cumulative incidence vary by assay; for example, lower RT-PCR cycle thresholds or the use of less sensitive rapid antigen tests would lead to lower estimates of cumulative cases. Finally, our analysis compares seroprevalence to cumulative case ratios at

different point in time. As diagnostic testing measures expanded, these ratios may have declined over time, complicating the process of applying a single fixed ratio to a cumulative incidence number. As such, there is a need for more nuanced analysis of case under-ascertainment and caution should be exercised if utilizing them in public health planning.

This study has limitations. Firstly, some asymptomatic individuals may not seroconvert, some individuals may have been tested prior to seroconversion, and others may have antibodies that have waned by the time of blood collection, so the data in this study may underestimate the number of SARS-CoV-2 infections [51]. To ameliorate this, we prioritized estimates that tested for anti-spike IgG antibodies, which show better persistence in serum compared to non-IgG and anti-nucleocapsid IgG antibodies [20–25]. Secondly, to account for measurement error in seroprevalence estimates resulting from poorly performing tests, it was necessary to use sensitivity and specificity information from multiple sources of varying quality. While we prioritized independent evaluations, these were not available for all tests. Furthermore, lab-to-lab variation may undermine the generalizability and comparability of the test evaluation data we utilized. Going forward, investigators should conduct evaluations of their assays using a standard international reference panel, such as the panel created by the WHO [52], and report their results in international units referenced against the World Antibody Titres Standard to increase comparability of serosurvey results. Where this is not feasible, investigators should at least report the test name, manufacturer, and sensitivity and specificity values to improve data comparability [53]. Thirdly, some of the summary results may have been driven by the large volume of data from high-income countries, which primarily reported lower seroprevalence estimates. While we frequently stratified by or adjusted for GBD region, caution is required when interpreting some of the summary estimates. Fourthly, the residual heterogeneity in our meta-regression indicates that not all relevant explanatory variables have been accounted for. Many factors may contribute to the spread of infection. Even if all important factors were known, it would be difficult to account for the variation in seroprevalence due to limited availability of data with sufficient granularity and changing health policy and individual behavior.

This systematic review is the largest synthesis of SARS-CoV-2 serosurveillance data to date. Our search was rigorous and comprehensive: we included non-English articles, government reports, unpublished data, and serosurveillance reports obtained via expert recommendations and the SeroTracker website. This comprehensive search is important because many serosurveys—especially in LMICs—have not been published or released as preprints. A strength of this review was the use of corrected prevalence estimates for analysis, revealing that imperfect sensitivity and specificity have major effects on seroprevalence findings. To our knowledge, this is the largest systematic comparison of seroprevalence estimates from blood donors, residual sera, and household and community-based general population samples. Finally, this study is part of a regularly-updated systematic review, and summary results will continue to be disseminated throughout the pandemic on a publicly available website (SeroTracker.com) [12].

Serosurveillance efforts so far have mostly taken the form of formal studies led by academic institutions. This approach makes sense when serosurveys are used as a tool to periodically monitor the spread of infection and identify high-risk groups. However, given the rise of more infectious SARS-CoV-2 variants, continued uncertainty about the global prevalence of infection, and variably quality of serosurvey design and reporting, more coordinated, standardized, and routine serosurveillance may be needed. Furthermore, as vaccines are deployed, there may be additional value derived from serosurveys, specifically in evaluating vaccine effectiveness in the real world, monitoring aggregate immunity arising from infection and vaccination, and measuring population antibody titres as a correlate of protection and as an indicator for vaccine boosters. Therefore, going forward, serosurveillance efforts may better serve end-users if they take the form of real-time monitoring programs housed in public health units, using

standardized serosurvey protocols and reporting. Leaders who can compare studies in their regions over time and pair vaccine distribution data with live serosurveys will be well-equipped to track the pandemic, understand the impact of variants, and monitor outcomes of vaccination efforts in their communities in real time.

## Conclusion

Our review shows that SARS-CoV-2 seroprevalence remains low in the general population, indicating the importance of remaining vigilant until vaccine-derived herd immunity is achieved. There are clear geographic and population differences in SARS-CoV-2 infection prevalence, with certain groups disproportionately affected. Policy and decision makers need to better protect these groups to reduce inequity in the impact of COVID-19.

As the COVID-19 pandemic progresses and serology data accumulate, ongoing evidence synthesis is needed to inform public health policy. We will continue to update our systematic review and seroprevalence dashboard to help address this need.

## Supporting information

**S1 Materials.**
(DOCX)

## Acknowledgments

We would like to thank Dr. Diane Lorenzetti, a health science librarian at the University of Calgary, for her assistance in developing the search strategies. We would like to thank Prof John Ioannidis for his suggestion to disaggregate the case to infection ratio by global burden of disease region given the under-representation of data from low and middle income countries. We would also like to thank all serosurvey authors who contributed data and enhanced the quality of this review. CPY and JP hold a "Chercheur-boursier clinicien" career award from the Fonds de recherche du Québec–Santé (FRQS). JC holds a Canada Research Chair in Global Environmental Health and Epidemiology.

## Author Contributions

**Conceptualization:** Niklas Bobrovitz, Rahul Krishan Arora, Tingting Yan, Timothy Grant Evans, Jesse Papenburg, Matthew P. Cheng.

**Data curation:** Niklas Bobrovitz, Rahul Krishan Arora, Christian Cao, Emily Boucher, Michael Liu, Claire Donnici, Mercedes Yanes-Lane, Sara Perlman-Arrow, Hannah Rahim, Natasha Ilincic, Mitchell Segal, Nathan Duarte, Jordan Van Wyk, Tingting Yan, Simona Rocco.

**Formal analysis:** Niklas Bobrovitz, Rahul Krishan Arora, Christian Cao, Emily Boucher, Michael Liu, Claire Donnici, Mercedes Yanes-Lane, Mairead Whelan, Sara Perlman-Arrow, Judy Chen, Hannah Rahim, Natasha Ilincic, Mitchell Segal, Nathan Duarte, Tingting Yan, Austin Atmaja, Abel Joseph, Lucas Penny, David A. Clifton.

**Funding acquisition:** Rahul Krishan Arora, Tingting Yan, Timothy Grant Evans.

**Investigation:** Niklas Bobrovitz, Rahul Krishan Arora, Christian Cao, Emily Boucher, Michael Liu, Claire Donnici, Mairead Whelan, Judy Chen, Hannah Rahim, Natasha Ilincic, Mitchell Segal, Nathan Duarte, Jordan Van Wyk, Tingting Yan, Austin Atmaja, Simona Rocco, Abel Joseph, Lucas Penny.

**Methodology:** Niklas Bobrovitz, Rahul Krishan Arora, Christian Cao, Emily Boucher, Michael Liu, Claire Donnici, Mercedes Yanes-Lane, Sara Perlman-Arrow, Hannah Rahim, Natasha Ilincic, Nathan Duarte, Jordan Van Wyk, Tingting Yan, David A. Clifton, Tyler Williamson, Cedric P. Yansouni, Timothy Grant Evans, Jonathan Chevrier, Jesse Papenburg, Matthew P. Cheng.

**Project administration:** Niklas Bobrovitz, Rahul Krishan Arora, Tingting Yan.

**Resources:** Rahul Krishan Arora, Tingting Yan.

**Software:** Rahul Krishan Arora, Michael Liu, Jordan Van Wyk, Austin Atmaja, Simona Rocco, Abel Joseph.

**Supervision:** Niklas Bobrovitz, David A. Clifton, Tyler Williamson, Cedric P. Yansouni, Timothy Grant Evans, Jonathan Chevrier, Jesse Papenburg, Matthew P. Cheng.

**Validation:** Niklas Bobrovitz, Rahul Krishan Arora, Christian Cao, Emily Boucher, Michael Liu, Claire Donnici, Hannah Rahim, Natasha Ilincic, Mitchell Segal, Nathan Duarte, Tingting Yan.

**Visualization:** Rahul Krishan Arora, Michael Liu, Jordan Van Wyk, Austin Atmaja, Simona Rocco, Abel Joseph.

**Writing – original draft:** Niklas Bobrovitz, Rahul Krishan Arora, Christian Cao, Emily Boucher, Michael Liu, Jesse Papenburg, Matthew P. Cheng.

**Writing – review & editing:** Niklas Bobrovitz, Rahul Krishan Arora, Christian Cao, Emily Boucher, Michael Liu, Claire Donnici, Mercedes Yanes-Lane, Mairead Whelan, Sara Perlman-Arrow, Judy Chen, Hannah Rahim, Natasha Ilincic, Mitchell Segal, Nathan Duarte, Jordan Van Wyk, Tingting Yan, Austin Atmaja, Simona Rocco, Abel Joseph, Lucas Penny, David A. Clifton, Tyler Williamson, Cedric P. Yansouni, Timothy Grant Evans, Jonathan Chevrier, Jesse Papenburg, Matthew P. Cheng.

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
