## [Decision Letter · Decision Letter 0]

1 Feb 2021

PONE-D-20-40466

Global seroprevalence of SARS-CoV-2 antibodies: a systematic review and meta-analysis

PLOS ONE

Dear Dr. Bobrovitz,

Thank you for submitting your manuscript to PLOS ONE. After careful consideration, we feel that it has merit but does not fully meet PLOS ONE’s publication criteria as it currently stands. Therefore, we invite you to submit a revised version of the manuscript that addresses the points raised during the review process.

We look forward to receiving your revised manuscript.

Kind regards,

Yury E Khudyakov, PhD

Academic Editor

PLOS ONE

Additional Editor Comments:

Your manuscript was reviewed by 2 experts in the field. Both reviewers provided suggestions to improve your paper. Please review the attached comments and provide your responses.

Journal Requirements:

2.) While we note that methods and results have been described in the Supplementary file, we would suggest that at least some of this information (for example, the search strategy, the list of studies included and the results of the risk of bias assessment) is reported in the main text.

3.) Please amend your list of authors on the manuscript to ensure that each author is linked to an affiliation. Authors’ affiliations should reflect the institution where the work was done (if authors moved subsequently, you can also list the new affiliation stating “current affiliation:….” as necessary).

4.) We note that Figure 2 and Appendix figure 1 in your submission contain map images which may be copyrighted. All PLOS content is published under the Creative Commons Attribution License (CC BY 4.0), which means that the manuscript, images, and Supporting Information files will be freely available online, and any third party is permitted to access, download, copy, distribute, and use these materials in any way, even commercially, with proper attribution. For these reasons, we cannot publish previously copyrighted maps or satellite images created using proprietary data, such as Google software (Google Maps, Street View, and Earth). For more information, see our copyright guidelines: http://journals.plos.org/plosone/s/licenses-and-copyright.

a.) You may seek permission from the original copyright holder of Figure 2 and Appendix figure 1 to publish the content specifically under the CC BY 4.0 license. 

b.) If you are unable to obtain permission from the original copyright holder to publish these figures under the CC BY 4.0 license or if the copyright holder’s requirements are incompatible with the CC BY 4.0 license, please either i) remove the figure or ii) supply a replacement figure that complies with the CC BY 4.0 license. Please check copyright information on all replacement figures and update the figure caption with source information. If applicable, please specify in the figure caption text when a figure is similar but not identical to the original image and is therefore for illustrative purposes only.

Reviewers' comments:

Reviewer's Responses to Questions

**Comments to the Author**

1. Is the manuscript technically sound, and do the data support the conclusions?

Reviewer #1: Yes

Reviewer #2: Yes

2. Has the statistical analysis been performed appropriately and rigorously? 

Reviewer #1: Yes

Reviewer #2: I Don't Know

3. Have the authors made all data underlying the findings in their manuscript fully available?

Reviewer #1: Yes

Reviewer #2: Yes

4. Is the manuscript presented in an intelligible fashion and written in standard English?

Reviewer #1: Yes

Reviewer #2: Yes

5. Review Comments to the Author

Reviewer #1: The manuscript summarises the serology studies available for SARS-CoV-2 through a systematic review and meta-analysis of the resulting data. In addition to describing the overall global seroprevalence, they characterise seroprevalence in various demographic groups e.g. age, gender, country, ethnicity, type of patient/sample etc. Since the approach taken corrects for bias such as lack of sensitivity/sensitivity or demographic adjustments in the original publications the results here should provide an increased accuracy in the estimates.

The manuscript is interesting and from what I understood from the methods, technically sound. However, it would benefit from some clarifications that I detail in the attached document.

Reviewer #2: This is a clear and well written report of a systematic review/meta-analysis of the literature on sera-prevalence of SARS-CoV-2 antibodies published worldwide. The authors have a clear understanding of the pitfalls associated with both study design and laboratory evaluation of population based seroprevalence and have brought together the world literature up to August 28th 2020 in an accessible way with appropriate corrections.

As this is such a rapidly evolving field, the only concern is whether this data adequately reflects the current situation. With a cut off date for analysis of late August 2020, most of the completed studies will represent seroprevalence estimates relatively early in the pandemic. If an updated analysis to the end of December 2020 could be incorporated into this manuscript that would be ideal and would add value as the authors could estimate seroprevalence in relation to time when the relevant population was sampled and that in turn could be evaluated in the context of time since the onset of the pandemic.

6. PLOS authors have the option to publish the peer review history of their article (what does this mean?). If published, this will include your full peer review and any attached files.

Reviewer #1: No

Reviewer #2: **Yes: **David Goldblatt

---

## [Author Response · Author response to Decision Letter 0]

16 May 2021

1. Responses to editor comments

1.1) Thank you for submitting your manuscript to PLOS ONE. After careful consideration, we feel that it has merit but does not fully meet PLOS ONE’s publication criteria as it currently stands. Therefore, we invite you to submit a revised version of the manuscript that addresses the points raised during the review process.

Thank you very much for reviewing our manuscript. We have responded to the editorial and reviewer comments below and resubmitted a revised version of the manuscript for your consideration. 

1.2) Please ensure that your manuscript meets PLOS ONE's style requirements, including those for file naming. The PLOS ONE style templates can be found at XX. 

Thank you. We have revised the style of the manuscript to meet PLOS ONE’s style requirements. 

1.3) While we note that methods and results have been described in the Supplementary file, we would suggest that at least some of this information (for example, the search strategy, the list of studies included and the results of the risk of bias assessment) is reported in the main text.

Thank you for this suggestion. We have moved the following information into the main text: search strategy, the summary risk of bias assessment figure, and Table 4 Summary of seroprevalence data for general and special population sub-groups (formerly appendix Table 4). 

1.4) Please amend your list of authors on the manuscript to ensure that each author is linked to an affiliation. Authors’ affiliations should reflect the institution where the work was done (if authors moved subsequently, you can also list the new affiliation stating “current affiliation:….” as necessary).

We have amended the list of authors to ensure that each is linked to an affiliation. 

1.5) We note that Figure 2 and Appendix figure 1 in your submission contain map images which may be copyrighted. All PLOS content is published under the Creative Commons Attribution License (CC BY 4.0), which means that the manuscript, images, and Supporting Information files will be freely available online, and any third party is permitted to access, download, copy, distribute, and use these materials in any way, even commercially, with proper attribution. For these reasons, we cannot publish previously copyrighted maps or satellite images created using proprietary data, such as Google software (Google Maps, Street View, and Earth). For more information, see our copyright guidelines: http://journals.plos.org/plosone/s/licenses-and-copyright.

a.) You may seek permission from the original copyright holder of Figure 2 and Appendix figure 1 to publish the content specifically under the CC BY 4.0 license. 

The revised version of this manuscript contains map figures that were generated using map data from Natural Earth (specific dataset at this link). Natural Earth provides map data that is in the public domain. 

From their About Page: “All versions of Natural Earth raster + vector map data found on this website are in the public domain. You may use the maps in any manner, including modifying the content and design, electronic dissemination, and offset printing. The primary authors, Tom Patterson and Nathaniel Vaughn Kelso, and all other contributors renounce all financial claim to the maps and invites you to use them for personal, educational, and commercial purposes. No permission is needed to use Natural Earth. Crediting the authors is unnecessary.”

In the caption of all map figures, we have added the following phrase: “Map data reprinted from Natural Earth under a CC BY license, with permission from Natural Earth, original copyright 2009.” If this phrase is unnecessary for public domain maps, we would also be glad for this to be removed. 

b.) If you are unable to obtain permission from the original copyright holder to publish these figures under the CC BY 4.0 license or if the copyright holder’s requirements are incompatible with the CC BY 4.0 license, please either i) remove the figure or ii) supply a replacement figure that complies with the CC BY 4.0 license. Please check copyright information on all replacement figures and update the figure caption with source information. If applicable, please specify in the figure caption text when a figure is similar but not identical to the original image and is therefore for illustrative purposes only.

USGS EROS (Earth Resources Observatory and Science (EROS) Center) (public domain): http://eros.usgs.gov/

1.6) Additional comments to the editor 

Note for the editor regarding Figure 1 and number of screened full texts: we re-classified our definition of a full text screen for Google News articles such that fewer articles were counted as full text in this updated review. While the overall number of full text articles screened has increased, a number of previous full text exclusions have been re-classified as abstract exclusions. The updated data are provided in the revised Figure 1. 

2) Reviewer 1 major comments: 

2.1) The manuscript summarises the serology studies available for SARS-CoV-2 through a systematic review and meta-analysis of the resulting data. In addition to describing the overall global seroprevalence, they characterise seroprevalence in various demographic groups e.g. age, gender, country, ethnicity, type of patient/sample etc. Since the approach taken corrects for bias such as lack of sensitivity/sensitivity or demographic adjustments in the original publications the results here should provide an increased accuracy in the estimate. The manuscript is interesting and from what I understood from the methods, technically sound. However, it would benefit from some clarifications that I detail below. 

Thank you for reviewing our article. We have addressed your requests for clarification below. 

2.2) The rationale for needing sero-surveys and a review of sero-surveys is not clear. For example, but not exclusively, in the introduction (3rd paragraph), the authors mention that is increasingly important to measure baseline prevalence of antibodies in the vaccine era. While I agree with importance of understanding serological patterns, sero tests are not being done prior to vaccination nor is being taken into account for the number of doses distributed since there is limited supply, and is also not being used for prioritizing groups. So why is it important that we know this? 

We have clarified the rationale for study as follows: 

Page 5, Line 105: Serological assays identify SARS-CoV-2 antibodies, indicating previous infection in unvaccinated persons.7 Population-based serological testing provides better estimates of the cumulative incidence of infection by complementing diagnostic testing of acute infection and helping to inform the public health response to COVID-19. Furthermore, as the world moves through the vaccine and variant era, synthesizing seroepidemiology findings is increasingly important to track the spread of infection, identify disproportionately affected groups, and measure progress towards herd immunity.

SARS-CoV-2 seroprevalence estimates are reported not only in published articles and preprints, but also in government and health institute reports, and media.8 Consequently, few studies have comprehensively synthesized seroprevalence findings that include all of these sources.9,10 Describing and evaluating the characteristics of seroprevalence studies conducted over the first year of the pandemic may provide valuable guidance for serosurvey investigators moving forward. 

2.3) Correction of seroprevalence estimates: This needs a bit more explanation and clarification throughout text and tables. For corrected seroprevalences, did you correct all studies or do you use a mixture of published corrections and corrections made in this study? Either way how do you ensure unbiased/equivalent corrections? 

We have provided more details about the correction of seroprevalence estimates in the methods section. This text clarifies that for corrected seroprevalences, we corrected all those studies where we had sufficient information to do so, even where published corrections were available; where we did not have sufficient informations, we used author-corrected seroprevalence estimates. 

Page 9-10, Line 200-208: To account for imperfect test sensitivity and specificity, seroprevalence estimates were corrected using Bayesian measurement error models, with binomial sensitivity and specificity distributions.26 The sensitivity and specificity values for correction were derived, in order of preference, from: (i) the FindDx -McGill database of independent evaluations of serological tests27; (ii) independent test evaluations conducted by serosurvey investigators and reported alongside serosurvey findings; (iii) manufacturer-reported sensitivity and specificity (including author evaluated in-house assays); (iv) published pooled sensitivity and specificity by immunoassay type.25 If uncorrected estimates were not available, we used author-reported corrected seroprevalence estimates. Details of these evaluations are located in S4 File. 

For the studies included in this meta-analysis, there was no comprehensive source of perfectly equivalent correction data available, where tests were evaluated against the same panel of samples and with the same antibody titre standard. These efforts are still underway by the WHO and other groups. For this reason, we have used data from several sources which use similar methods to evaluate these tests. While this method cannot completely remove bias related to test performance, it minimizes bias as compared to not correcting for test sensitivity and specificity altogether.

We have provided more details in the results section about the sources of correction that were available for the final data set. 

Page 14-15, Line 277-300: In order to improve comparability between data and correct for misclassification error, we corrected seroprevalence values for imperfect sensitivity and specificity. To do so, we sourced additional evaluation data as described in the methods. Overall, there were 795 studies (82%) for which test sensitivity and specificity values were reported or located (S5 Table). Authors reported sensitivity and specificity data in 229 studies, with reported sensitivity values ranging from 35-100% and specificity between 87-100%.

Independent evaluation data from the FindDx initiative were available for 359 studies (37%), manufacturer evaluations were available for 182 studies (19%), and published pooled sensitivity and specificity results for ELISAs, LFIAs, and CLIAs, based on the test type known to have been used, and using the definitions for these test types provided by Bastos et al.25, were available for 101 studies (10%). Between FindDx, manufacturer evaluations, and published pooled results, test sensitivity ranged from 9-100% and specificity from 0-100%. 

Estimates from 587 studies (61%) were corrected for imperfect sensitivity and specificity. We corrected seroprevalence estimates from 290 studies (30%), while author-corrected estimates were used in 297 (31%) studies as uncorrected estimates were not available for our analysis. The median absolute difference between corrected and uncorrected seroprevalence estimates was 1.1% (IQR 0.6-2.3%). 

Of the 381 studies for which estimates were not corrected, data were insufficient to inform the correction analysis in 118 studies (12%). Corrected seroprevalence estimates could not be determined for 261 studies (27%), most of which were population-specific studies using small sample sizes and low test sensitivity and specificity. In these studies, the model used to correct for test sensitivity and specificity often failed to converge to a reasonable adjusted prevalence value.

2.4) What sort of independent evaluations did you base your corrections, are these for sensitivity/specificity values provided by commercial kits? 

We have provided more details of the source of independent evaluations in the main text and highlighted the additional information on these sources in the appendix. Most of these independent evaluations were for sensitivity and specificity in commercial kits. 

Page 9-10, Line 200-208: To account for imperfect test sensitivity and specificity, seroprevalence estimates were corrected using Bayesian measurement error models, with binomial sensitivity and specificity distributions.26 The sensitivity and specificity values for correction were derived, in order of preference, from: (i) the FindDx -McGill database of independent evaluations of serological tests27; (ii) independent test evaluations conducted by serosurvey investigators and reported alongside serosurvey findings; (iii) manufacturer-reported sensitivity and specificity (including author evaluated in-house assays); (iv) published pooled sensitivity and specificity by immunoassay type.25 

S4 file, page 11, line 171-206: The sensitivity and specificity values for correction were derived, in order of preference, from: (i) the FINDDx-McGill database of independent evaluations of serological tests9; (ii) independent test evaluations conducted by serosurvey investigators and reported alongside serosurvey findings; (iii) manufacturer-reported sensitivity and specificity; (iv) published pooled sensitivity and specificity by immunoassay type.10 If uncorrected estimates were unavailable, we used author-reported corrected seroprevalence estimates in lieu of performing our own correction. When none of the above corrections were possible, we excluded estimates from further analysis. Details of this order of priority follow:

1. The FINDDx-McGill database of independent evaluations of serological tests.9 We only considered evaluations reporting both sensitivity and specificity for test performance across all sickness days (as opposed to day 1, day 5, etc). Where multiple evaluations were available, we prioritized in the following order: 

a. The evaluation needed to match the test name, manufacturer, and target isotype used in the study 

b. The evaluation needed to match the sample specimen type used in the study

i. For sample types that were not reported in either the test evaluation or the serosurvey study we assumed whole blood was used for LFIA tests and serum/plasma was used for non-LFIA tests

ii. Plasma/Serum were used interchangeable when no direct match for index sample type was available 

c. Prioritized reference specimen type which yield the most virus according to a systematic review and meta-analysis published in July 2020 that compared RT-PCR positivity of different specimens11

i. It was assumed that “respiratory specimen” was referring to upper respiratory specimen as a conservative assumption as these viral loads are lower than bronchoalveolar lavage or sputum. We ranked it along side throat swab. 

ii. “Lower respiratory specimen” was ranked with with bronchoalveolar lavage fluid 

iii. “Upper respiratory specimen” was ranked with throat swab

iv. If a mixed reference sample was used in the independent evaluation then an even distribution of sample types was assumed; the average % yield of the viral load was calculated and the sample type was ranked accordingly 

d. Largest sample size

2. Serological test evaluations conducted by study authors, where those authors were at arms-length from the design of the study in questions

3. Manufacturer-reported sensitivity and specificity, which includes evaluations of in-house serological tests published by the research group that developed the same test

4. Published pooled sensitivity and specificity results for ELISAs, LFIAs, and CLIAs, based on the test type known to have been used, and using the definitions for these test types provided in the cited article.10 

2.5) What about lab-to-lab variations and in-house assays? 

Excellent question. Lab to lab variation is a major barrier to comparisons of assay evaluations and serological study findings. We have added a statement about this to the discussion section and highlight the need for use of international standards such as the WHO standard reference panel and international antibody titre standard described in January 2021. Data using this standard will begin to become more broadly available later in 2021. 

Page 27-28, line 484-494: Secondly, to account for measurement error in seroprevalence estimates resulting from poorly performing tests, it was necessary to use sensitivity and specificity information from multiple sources of varying quality. While we prioritized independent evaluations, these were not available for all tests. Furthermore, lab-to-lab variation may undermine the generalizability and comparability of the test evaluation data we utilized. Going forward, investigators should conduct evaluations of their assays using a standard international reference panel, such as the panel created by the WHO52, and report their results in international units referenced against the World Antibody Titres Standard to increase comparability of serosurvey results. Where this is not feasible, investigators should at least report the test name, manufacturer, and sensitivity and specificity values to improve data comparability.53

If in-house assays designed by the authors of the seroprevalence study were used, and the authors reported sensitivity and specificity values from in-house validation, these test evaluations were considered to be manufacturer values (as they were done by the same group developing the assay) and used to independently correct seroprevalence estimates. We have added a statement to the methods and appendix to clarify this. 

Page 9-10, Line 200-208: To account for imperfect test sensitivity and specificity, seroprevalence estimates were corrected using Bayesian measurement error models, with binomial sensitivity and specificity distributions.26 The sensitivity and specificity values for correction were derived, in order of preference, from: (i) the FindDx -McGill database of independent evaluations of serological tests27; (ii) independent test evaluations conducted by serosurvey investigators and reported alongside serosurvey findings; (iii) manufacturer-reported sensitivity and specificity (including author evaluated in-house assays); (iv) published pooled sensitivity and specificity by immunoassay type.25

Page 12, line 203-204: Manufacturer-reported sensitivity and specificity, which includes evaluations of in-house serological tests published by the research group that developed the same test

2.6) What type of sensitivity analysis was conducted on uncorrected data and for what? 

We reported the uncorrected data for all major analyses in the main text and/or the appendix so that readers can directly compare the results. The following data displays include uncorrected data:

Table 2. Summary of seroprevalence data from studies reporting population-wide estimates by global burden of disease region, geographic scope, and risk of bias

Table 3. Summary of seroprevalence data by study sampling frame 

Table 4. Summary of seroprevalence data from studies reporting population-specific estimates by global burden of disease region, geographic scope, and risk of bias

S4 Table. Summary of unadjusted meta-analysis results

S6 Table. Summary of meta-regression results

In the results section, we also present the median absolute difference between corrected and uncorrected seroprevalence estimates for population-wide studies. 

Results, page X, Line X-X: The median absolute difference between corrected and uncorrected seroprevalence estimates was 1.1% (IQR 0.6-2.3%).

2.7) How do you correct for power of the studies? The sample size of studies would have been planned taking into account the population size and demographic of the region, hence providing a powered measure of seroprevalence, but many not. 

Our Risk of Bias assessments contain an item which accounts for study power. Item 3 in the Risk of Bias tool evaluates study sample size. Our scoping work on this topic revealed that very few studies (< 10%) reported sample size calculations. For this reason, we carried out our own calculations to determine a threshold sample size: n = 599, which is sufficient to have 80% power in detecting a 2.5% seroprevalence to a precision of 1.5%. The full rationale for this and details have been added to the S3 file. 

S3 file, page 8, line 99: To calculate the required sample size we used an assumed prevalence of 2.5%, which was the global average estimated by the WHO in April, 2020.3 Based on guidance by the Joanna Briggs Institute and published medical statistical recommendations we selected a precision value that was half the assumed prevalence (1.25%)4,5 We calculated a minimum sample size of 599 using these inputs:

Sample size calculation: 

Where n = sample size;

Z = Z statistic for level of confidence (95%);

P = expected prevalence (2.5% WHO global estimate);

d = precision (1.25%)

In cases where the sample size calculation was provided, this item was marked as yes — even if the required sample for 80% power was below the n = 599 threshold. 

2.8) How did the meta-analysis account for the level of risk of bias identified? And how can we interpret this risk of bias? 

Both our meta-regression and meta-analysis of seroprevalence ratios accounted for study risk of bias.

First, we conducted a multivariable linear meta-regression which included risk of bias as a categorical covariate. This analysis is described in the methods. 

Page 10, line 214-219: To examine study-level factors affecting population-wide seroprevalence estimates, we constructed a multivariable linear meta-regression model. The outcome variable was the natural logarithm of corrected seroprevalence. Independent predictors were defined a priori. Categorical covariates were encoded as indicator variables, and included: study risk of bias (reference: low risk of bias), GBD region (reference: high-income); geographic scope (reference: national); and population sampled (reference: household and community samples).

Second, we conducted a meta-analysis to identify differences between sub-groups within studies. This analysis is described in the methods. In this analysis, the seroprevalence ratio between groups (e.g., the ratio between the seroprevalence in males and the seroprevalence in females) was first calculated within each study, so the risk of bias was controlled for inherently. The ratios were then pooled across studies. 

Page 11, line 227-232: To quantify population differences in SARS-CoV-2 seroprevalence, we identified subgroup estimates within population-wide studies that stratified by sex/gender, race/ethnicity, contact with individuals with COVID-19, occupation, and age groups. We calculated the ratio in prevalence between groups within each study (e.g., prevalence in males vs. females) then aggregated the ratios across studies using inverse variance-weighted random-effects meta-analysis (S4 File). Heterogeneity was quantified using the I² statistic.35

We provide an interpretation of the risk of bias in the Supplement. The definition centres on the degree to which there is systematic error in an estimate that would result in its deviation away from the “true” value. 

S3 file, page 10, line 120: 

Item 10: Risk of bias

Low The estimates are very likely correct for the target population. To obtain a low risk of bias classification, all criteria must be met or departures from the criteria must be minimal and unlikely to impact on the validity and reliability of the prevalence estimate. These include sample sizes that are just below the threshold when all other criteria are met, reporting only some of characteristics of the sample, test characteristics below the threshold but corrections for the test performance, and response rates that are just below the threshold in the context of probability based sampling of an appropriate sampling frame with population weighted seroprevalence estimates.

Moderate The estimates are likely correct for the target population. To obtain a moderate risk of bias classification, most criteria must be met and departures from the criteria are likely to have only a small impact on the validity and reliability of the prevalence estimates.

High The estimates are not likely correct for the target population. To obtain a high risk of bias, many criteria must not be met or departures from criteria are likely to have a major impact on the validity and reliability of the prevalence estimates.

Unclear There was insufficient information to assess the risk of bias.

 

Reviewer 1 minor comments: 

2.9) Introduction: 2nd paragraph: ‘previous infection’ – infection or exposure? 

Previous infection was intended here, as individuals who were exposed may not be infected and may not mount an immune response. 

Page 5, line 105: Serological assays identify SARS-CoV-2 antibodies, indicating previous infection in unvaccinated persons.7

2.10) Introduction: 4th paragraph: what gap? There was no clear gap identified up to here. 

We have removed this language. 

Page 6, line 118: We conducted a systematic review and meta-analysis of SARS-CoV-2 seroprevalence studies published in 2020

2.11) Introduction- what is the start and end dates for the lit review? 

We have added the date

Page 6, line 118: We conducted a systematic review and meta-analysis of SARS-CoV-2 seroprevalence studies published in 2020.

These details are also provided in the methods. 

Page 7, Line 149: Our search dates were from January 1, 2020 to December 31, 2020.

2.12) Introduction- ‘true burden’ – I wonder if this is the best term (which is mentioned throughout the manuscript). Doesn’t burden refer to mortality and morbidity? Or at least something that incurs some sort of cost. Many, if not most, seropositives will have been asymptomatic. 

Throughout the manuscript we have replaced the term “burden” with “spread”, “infection”, or “prevalence”. 

2.13) Data sources: Is there a reason to exclude PubMed? 

Our health sciences librarian, Diane Lorenzetti, advised us that 98% of articles in PubMed are captured in the MEDLINE database. Given the large scope of our search strategy (4 databases, 4 public health agency websites, Google News search, serotracker platform submissions, expert recommendations) and ongoing nature of the living review we have tried to balance comprehensiveness with feasibility. 

2.14) Data sources: Who is the librarian? At least add the affiliation. 

Our librarian is Diane Lorenzetti. We have acknowledged her in the manuscript and have now added her affiliation to this acknowledgment. 

Page 30, line 539-540: We would like to thank Dr. Diane Lorenzetti, a health science librarian at the University of Calgary, for her assistance in developing the search strategies.

2.15) Data sources: key eligibility criteria/ search words should be specified in the main text. 

We have added details on key eligibility criteria to the methods section of the main text.

Page 7-8, line 154-168: We included SARS-CoV-2 serosurveys in humans. We defined a single serosurvey as the serological testing of a defined population over a specified time period to estimate the prevalence of SARS-CoV-2 antibodies.14,15 To be included, studies had to report a sample size, sampling date, geographic location of sampling, and prevalence estimate. Articles not in English or French were included if they could be fully extracted using machine translation.16 Articles that provided information on two or more distinct cohorts (different sample frames or different samples at different time points) without a pooled estimate were considered to be multiple studies. 

If multiple articles provided unique information about a study, both were included. Articles reporting identical information to previously included articles were excluded as duplicates – this rule extended to pre-print articles that were subsequently published are peer-reviewed journals. In these cases, the peer-reviewed articles were considered the definitive version. 

We have added details on the search to the methods section of the main text. The search strategies themselves are extensive; for this reason, we have left them in the supplement. 

Page 6, line 129-135: We searched Medline, EMBASE, Web of Science, and Europe PMC, using a search strategy developed in consultation with a health sciences librarian (DL). The strategies for MEDLINE and EMBASE were an expanded version of the published COVID-19 search strategies created by OVID librarians for these databases.13 Search terms related to serologic testing were identified by infectious disease specialists (MC, CY, and JP)7 and expanded using Medical Subject Heading (MeSH) or Emtree thesauri. These searches were adapted for the other databases. The full search strategy can be found in S2 File.

2.16) Study selection: ‘SARS-CoV-2 infection’ –do you mean studies that included only previously PCR positives? 

We have revised this exclusion criteria statement to offer more clarity. 

Page 8, line 165-168: We excluded studies conducted only in people previously diagnosed with COVID-19 using PCR, antigen testing, clinical assessment, or self-assessment; dashboards that were not associated with a defined serology study; and case reports, case-control studies, randomized controlled trials, and reviews. 

2.17) Study selection: Associated factors: there are far more studies for high-income countries, how do you take study effort into account for global or even large regional scales? 

We have stratified the results by Global Burden of Disease region so that readers are aware of the proportion of data coming from high-income countries. In the meta-regression, we included study Global Burden of Disease region as a categorical covariate. We highlight in the discussion that the majority of data comes from high-income countries and that some of the estimates may therefore be driven by these data. We recommend that more studies be conducted in low and middle income countries. 

Page 28, line 494-497: Thirdly, some of the summary results may have been driven by the large volume of data from high-income countries, which primarily reported lower seroprevalence estimates. While we frequently stratified by or adjusted for GBD region, caution is required when interpreting some of the summary estimates.

2.18) Results: what is considered general and special populations? 

For clarity, we have changed this terminology to studies providing either population-wide or population-specific estimates. We have provided definitions for these groups in the methods. 

Page 9, line 188-192: Seroprevalence studies were grouped as providing either population-wide or population-specific estimates. Population-wide studies included those using household or community sampling frames as well as convenience samples from blood donors or residual sera used for monitoring other conditions in the population. Population-specific studies were those sampling from well-defined population sub-groups, such as health care workers or long-term care residents. 

2.19) Results: blood donors seem to be considered as general population, given they are typically young and healthier/fiter than average, are they not a special population? 

Given that public health agencies often use blood donor samples as a practical strategy to measure seroprevalence in general population, we have opted to categorize them as studies providing population-wide seroprevalence estimates. We acknowledge the demographic and behavior differences between blood donors and the broader community and cite this in our discussion. Our meta-regression also quantifies the difference between seroprevalence in household/community samples, blood donor samples, and residual sera samples. After adjusting for confounding factors, the results show no statistically significant difference. This is a useful seroepidemiological finding that we have added to the discussion. 

Page 15, line 302-306: In studies reporting population-wide seroprevalence estimates, median corrected seroprevalence was 4.5% (IQR 2.4-8.4%, Table 2). These studies included household and community samples (n=125), residual sera (n=248), and blood donors (n=54), with median corrected seroprevalence of 6.0% (IQR 2.8-15.1%), 4.0% (IQR 2.4-6.8%), and 4.7% (IQR 1.4-6.8%), respectively (Table 3). 

Page 24-25, line 415-425: Approximately half of studies reporting population-wide SARS-CoV-2 seroprevalence estimates used blood from donors and residual sera as a proxy for the community. Our results showed that these studies report seroprevalence estimates that are similar to studies of household and community-based samples. It has previously been shown that these groups contain disproportionate numbers of people that are young, White, college graduates, employed, physically active, and never-smokers.47,48 However, the results of our study suggest that investigators may use these proxy sampling frames to obtain fairly representative estimates of seroprevalence if studies use large sample sizes with adequate coverage of important subgroups (e.g., age, sex, race/ethnicity) to permit standardization to population characteristics, tests with high sensitivity and specificity, and statistical corrections for imperfect sensitivity and specificity. 

2.20) Results: the time window for these estimates need to be stated at the start of the results. I would imagine that now, seroprevalence is considerably higher in many regions/groups. 

We have added this date range to the start of the results. 

Page 12, line 250-251: Study sampling dates ranged from September 1, 2019 to December 31, 2020.

2.21) Table 4: Could remove rows for reference as this information is already in columns. The risk seems higher for children than adults? This seem to contradict many studies no? 

Thank you for this suggestion. We have removed the reference rows from Table 5. 

Using updated data, the results show that the risk for adults and children are not significantly different. 

2.22) I wonder if some of the tables can be transformed into plots for an easier visualization? 

We have included two additional figures in the main text to help with visualization (Figure 2, Figure 3). 

2.23) Conclusion: 2nd paragraph: Or baseline health…. The sentence starting ‘Given’ is important and should be expanded. How does Community transmission impact SARS-CoV-2 transmission? It currently read transmission impacts transmission which seems a bit circular and empty. Is community transmission a proxy or behaviour? 

Thank you for pointing this out. We have revised this statement in the conclusion. 

Page 23, line 384-387: Given the limited evidence for altitude or climate effects on SARS-CoV-2 transmission36,37 variations in seroprevalence likely reflect differences in community transmission based on behaviour, public health responses, local resources, and the built environment.

2.24) Conclusion: what are the units of (24.0 local vs 11.9 national vs 15.7 regional)? 

These were ratios between seroprevalence and cumulative incidence. We have clarified this metric in the conclusion. 

Page 26, line 449-451: Seroprevalence estimates were 18.1 times higher than the corresponding cumulative incidence of COVID-19 infections, with large variations between the Global Burden of Disease Regions (seroprevalence estimates ranging from 6 to 602 times higher than cumulative incidence).

2.25) Conclusion: the 11.9 ratio values is without applying spatial heterogeneity in under-ascertains both between countries and within a country - and is biased by the countries that had capacity to perform a serological test. How would these estimates change if these heterogeneities were included? 

This is a very important point. Thank you for raising it. We have provided a stratified analysis showing how the ratio between seroprevalence and cumulative incidence varies by Global Burden of Disease region. This means that separate estimates are now provided for high-income countries globally and for the low- and middle-income countries in each World Health Organization region. We now comment on these issues in the discussion and highlight the limitations of this data. We agree that bias in the overall estimate is introduced based on the disproportionate amount of data coming from high income countries and caution readers about this. 

Page 26-27, line 449-478: Seroprevalence estimates were 18.1 times higher than the corresponding cumulative incidence of COVID-19 infections, with large variations between the Global Burden of Disease Regions (seroprevalence estimates ranging from 6 to 602 times higher than cumulative incidence). This level of under-ascertainment suggests that confirmed SARS-CoV-2 infections are a poor indicator of the extent of infection spread, even in high-income countries where testing has been more widely available. The broad range of ratios mirrors estimates from other published evidence on case under-ascertainment, which suggests a range of 0.56 to 717.49,50

Seroprevalence to cumulative case ratios can provide a rough roadmap for public health authorities by identifying areas that may be receiving potentially insufficient levels of testing and by providing an indication of the number of undetected asymptomatic infections.

While there is interest in using these seroprevalence to cumulative case ratios in identifying inadequate testing and estimating case ascertainment, caution is required in the quantitative interpretation of these ratios. Our study found a median ratio of 18.1, which aligns with other published analysis.50 This would imply that 2.9 billion people globally have been infected with SARS-CoV-2 rather than the 160 million reported as of May 15, 2021.2 This is not likely, and this estimate conflicts with the evidence that seroprevalence remains low in the general population. If applying this global ratio to countries with high cumulative incidence, such as the United States (32 million by May 15, 2021), then the total number of infections would exceed the population.

There are several possible reasons for these discrepancies. Firstly, these ratios clearly vary by geographic region and regional health policy, with higher diagnostic testing rates likely to correspond to lower seroprevalence to case ratios. Country-specific ratios, or region-specific ratios if available, should be used to inform planning wherever possible. Second, diagnostic testing-based estimates of cumulative incidence vary by assay; for example, lower RT-PCR cycle thresholds or the use of less sensitive rapid antigen tests would lead to lower estimates of cumulative cases. Finally, our analysis compares seroprevalence to cumulative case ratios at different point in time. As diagnostic testing measures expanded, these ratios may have declined over time, complicating the process of applying a single fixed ratio to a cumulative incidence number. As such, there is a need for more nuanced analysis of case under-ascertainment and caution should be exercised if utilizing them in public health planning.

 

2.26) Conclusion: P17 1st parag: ‘may not seroconvert’ - or antibodies could have wained by the time of blood collection... 

Thank you for highlighting this. We have added that statement to the conclusion section. 

Page 27, line 479-482: Firstly, some asymptomatic individuals may not seroconvert, some individuals may have been tested prior to seroconversion, and others may have antibodies that have waned by the time of blood collection, so the data in this study may underestimate the number of SARS-CoV-2 infections.51

2.27) Conclusion: Many studies have repeated patients. Was this considered? 

This was considered. Part of our review process includes identifying studies with overlapping participants and linking the studies in our database. As such, these participants are not double counted during analysis. 

2.27) Conclusion: P17 2nd parag: ‘there may be other factors…’ such as what? 

We have provided more information about this limitation and added examples of potential confounding factors. 

Conclusion, page X, line X-X: Fourthly, the residual heterogeneity in our meta-regression indicates that not all relevant explanatory variables have been accounted for. Many factors may contribute to the spread of infection. Even if all important factors were known, it would be difficult to account for the variation in seroprevalence due to limited availability of data with sufficient granularity and changing health policy and individual behavior.

2.28) Conclusion: P17 3rd parag: given the different level of scrutiny of these types of articles, do you think the results are comparable? 

We agree that these different articles are subject to varying levels of scrutiny. We have relied on the risk of bias checklist and meta-regression to increase the comparability of these articles. 

 

3) Reviewer 2 comments:

3.1) Reviewer #2: This is a clear and well written report of a systematic review/meta-analysis of the literature on sera-prevalence of SARS-CoV-2 antibodies published worldwide. The authors have a clear understanding of the pitfalls associated with both study design and laboratory evaluation of population based seroprevalence and have brought together the world literature up to August 28th 2020 in an accessible way with appropriate corrections.

Thank you for reviewing our manuscript. We have responded to your request below. 

3.2) As this is such a rapidly evolving field, the only concern is whether this data adequately reflects the current situation. With a cut off date for analysis of late August 2020, most of the completed studies will represent seroprevalence estimates relatively early in the pandemic. If an updated analysis to the end of December 2020 could be incorporated into this manuscript that would be ideal and would add value as the authors could estimate seroprevalence in relation to time when the relevant population was sampled and that in turn could be evaluated in the context of time since the onset of the pandemic.

Thank you for suggesting this. We agree that data is rapidly emerging. As such, our team of reviewers have updated the search to include literature from January 1, 2020 to December 31, 2020. The review is now triple the size of the original draft. It grew from 338 studies reported in 221 articles to 968 studies reported in 605 articles. We have positioned this review as a summary of seroprevalence studies in 2020. 

As the pandemic developed at different rates in different locations we have included a variable in the analysis to account for cumulative incidence of cases and, therefore, time when the relevant population was sampled relative to the onset of the pandemic in each country.

---

## [Editor Report · Decision Letter 1]

19 May 2021

Global seroprevalence of SARS-CoV-2 antibodies: a systematic review and meta-analysis

PONE-D-20-40466R1

Dear Dr. Bobrovitz,

We’re pleased to inform you that your manuscript has been judged scientifically suitable for publication and will be formally accepted for publication once it meets all outstanding technical requirements.

Kind regards,

Yury E Khudyakov, PhD

Academic Editor

PLOS ONE
---

## [Editor Report · Acceptance letter]

15 Jun 2021

PONE-D-20-40466R1 

Global seroprevalence of SARS-CoV-2 antibodies: a systematic review and meta-analysis 

Dear Dr. Bobrovitz:

I'm pleased to inform you that your manuscript has been deemed suitable for publication in PLOS ONE. Congratulations! Your manuscript is now with our production department. 

Kind regards, 

on behalf of

Dr. Yury E Khudyakov 

Academic Editor

PLOS ONE